# Headache Attributed to SARS-CoV-2 Infection or COVID-19 Related Headache—Not Migraine-like Problem-Original Research

**DOI:** 10.3390/brainsci11111406

**Published:** 2021-10-25

**Authors:** Aleksandra Kacprzak, Daniel Malczewski, Izabela Domitrz

**Affiliations:** 1Bielański Hospital, ul. Cegłowska 80, 01-809 Warszawa, Poland; 2Department of Neurology, Faculty of Medical Sciences, Medical University of Warsaw, 01-809 Warszawa, Poland; daniel.malczewski@wum.edu.pl (D.M.); izabela.domitrz@wum.edu.pl (I.D.)

**Keywords:** headache, migraine, COVID-19, SARS-CoV-2

## Abstract

Background: Many studies have confirmed headache as one of the most common COVID-19-related neurological symptoms. There are some reports concerning migraine attacks during SARS-CoV-2 infection with an unusual course of migraine attack. Our aim was to recognize and characterize accurately the features of headaches accompanying this disease. Methods: Research based on questionnaire study gathered 100 randomly chosen medical healthcare employees who experienced symptoms associated with COVID-19 disease, 96 with confirmed COVID-19 (positive SARS-CoV-2 PCR laboratory test or positive rapid COVID-19 antigen test). Conclusion: Headaches reported in the study did not fulfill criteria for migraine with/without aura, tension-type headache according to ICHD-3.

## 1. Introduction

The first cases of unknown severe pneumonia were observed in Wuhan, the capital of Hubei Province, China, in December 2019. The WHO officially named the disease Coronavirus Disease 2019 (COVID-19), and declared it as pandemic on 11 March 2020. Over the years, several different respiratory viruses have been shown to be able to penetrate the CNS (neuroinvasion). Most of them can infect neurons and glial cells (neurotropism) and result in the induction of neurological diseases [1]. However, it has now been proven indubitably that coronaviruses possess neurotropic and neuroinvasive properties in various hosts including humans, rats, pigs, rodents and fowl [2]. Coronaviruses first target respiratory and mucosal surfaces and then, depending on the host and virus strain, may spread to other tissues (brain, eyes, liver, kidneys and spleen) and cause a range of pathologies such as pneumonia, encephalitis, neurodegenerative demyelination, hepatitis, enteritis, and nephritis among others [3]. Many neurological manifestations have been reported in the literature associated with COVID-19, which we can classify into central nervous system (CNS)-related manifestations including headache, dizziness, impaired consciousness, acute cerebrovascular disease, and epilepsy, or peripheral nervous system (PNS) related manifestations such as hyposmia/anosmia, hypogeusia/ageusia, muscle pain, and Guillain–Barre syndrome [4].

N. Kaur et al., in their study, aimed to perform a systematic review of the current published literature on COVID-19 to provide insight on the epidemiological and clinical characteristics of COVID-19 patients. The study was a pooled analysis 6635 patients from 50 articles published until April 2020. The authors indicate that the prevalence of headache in patients with COVID-19 was around 12% [5]. Another early 2020 study focused on general COVID-19 symptom characterization calculated the prevalence of headache at slightly more than 10% (10.9%, 95%CI 8.62–13.51%) [6].

We created our study to focus on primary headache disorders and its comparison to headache features in COVID-19 and compared the results of other researchers, in order to diffuse and popularize that issue, making diagnostics easier to try to categorize this new medical problem. The aim of our research was not to find statistical correlation between headache and other symptoms in COVID-19, but attempt to characterize types of headaches in coronavirus infections. This is a trial to gathering all research participant complaints during their coronavirus infection, comparing them with other well-known and classified headaches and also on secondary headaches connected with ARS, to categorize headaches related to SARS-CoV-2 infection.

## 2. Methods

### 2.1. Study Population and Eligibility

The volunteers who participated in the survey were randomly chosen patients and medical healthcare employees from one of the largest hospitals in Warsaw. Participants were asked for permission to take part in the questionnaire study and all accepted and answered our questions. No one was forced to take part in the questionnaire. We created our questionnaire to focus on primary headache disorder and its comparison to headache features in COVID-19. Data were gathered between 20 December 20 2020 and 1 February 2021. The study included only participants who suffered from COVID-19 disease in last 3 months and with a confirmed COVID-19 test (2 possibilities of confirmation: positive SARS-CoV-2 PCR laboratory test or positive rapid COVID-19 antigen test). In the investigation were included adult participants from 18 to 70 years old. Surveys completed by participants under the age of 18, with past migraine headache history or reporting inconsistent responses were excluded. Average age was 39.3 SD. The study gathered 100 volunteers with confirmed COVID-19 by tests: 89 participants with COVID-19 confirmed by PCR laboratory tests, and 20 participants confirmed by rapid COVID-19 antigen tests. Some of the participants had confirmation from both tests. Predominantly women participated in the study (54 women, 46 men). The mean age of women and men was 39.3. None of our participants needed hospitalization because of COVID-19 disease.

### 2.2. Data Collection

A headache specialist designed a questionnaire containing the main semiological aspects of headaches. The questionnaire included questions of headache features such as location (bilateral, unilateral, partial), quality (tension, pulsation), duration of pain (permanent, episodic, duration and frequency of singular headache episodes), intensity of pain, associated symptoms like nausea, vomiting, hypersensitivity to light and sound, and headache connection with physical effort. Participants were also asked about information on non-headache related COVID-19 symptoms, sex and age, presence of fever during disease, headache treatment response and medications used. Questions were also included about the period leading up to headache, i.e., presence of flickering scotoma, vision disturbances. Volunteers had no history of migraine headaches, but some of them admitted intermittent tension-type headache occurrence.

## 3. Results

The disease can range from mild with asymptomatic or non-specific symptoms to severe with respiratory distress, as in Table 1. Participants most frequently reported headache, fever, skeletal muscle and joint pain, and feelings of drowsiness or weakness in COVID-19 disease.

In our study group, headache was mostly described as a tension-type headache, among 36/83 (43%) participants. The most frequent headache localizations reported were: whole head 49%, frontal lobe 29%, occipital lobe 10% and unilateral pain localization 6%. Study volunteers mostly evaluated headache intensity from middle to severe. The mean headache episode lasted 7 h 53 min. Accurate headache characteristics responded by participants are presented in Table 2.

Headache was the most frequently reported symptom in COVID-19 disease, admitted by 83% study volunteers. It was accompanied by skeletal muscle and joint pain in 50% of respondents, and feelings of drowsiness or fatigue in 49%. Fever, conventionally defined as body temperature above 38 degrees, was observed in 60% of participants. Headache in feverish persons affected 57/60 (95%), which is not a surprising outcome, when we take into consideration typical fever and headache in acute upper respiratory tract viral infection concomitance. On the other hand, headaches in persons who did not admit fever, affected 26/40 (65%), indicating headaches had a high independent occurrence. Other often-reported i symptoms present in most COVID-19 related studies worldwide, were cough, anosmia/hyposmia, ageusia/dysgeusia, and gastrointestinal disturbances.

## 4. Discussion

In initial studies from China the most common described symptoms of SARS-CoV-2 infection were: fever, cough, fatigue/myalgia, sputum, and dyspnea along with other clinical symptoms [7,8]. In the study of Zou et al., the main prevalent symptoms of hospitalized patients were fever (94%) and cough (79%) [9]. One European study focused on epidemiological characteristics of 1420 patients hospitalized with COVID-19 reported occurrence of headache in 998 patients (70%). In this study the most prevalent symptoms were headache (70.3%), loss of smell (70.2%), nasal obstruction (67.8%), cough (63.2%), asthenia (63.3%), myalgia (62.5%), rhinorrhea (60.1%), gustatory dysfunction (54.2%) and sore throat (52.9%). Fever itself was reported by 45.4% [10].

In many studies of the neurological manifestation of COVID-19, a noticeably higher prevalence of headaches accompany coronavirus disease, which is approximate to our study outcome. In research including 73 patients Rocha-Filho et al. analysed the association between headache and anosmia and ageusia. Forty-seven patients (64%) admitted headache occurrence in COVID-19. Bilateral headache localization was reported by 94% of patients, headache severity was determined as severe in 53%, and constant headaches with median period 15 days occurred in 11 of the 73 patients (15%). Twenty-four patients admitted pulsatile headache quality (51%) and photophobia occurred in 21 patients, phonophobia 14 patients. Researchers concluded that headache presents a migraine phenotype in 24 of patients (51%), a TTH phenotype in 19 patients (40%) and is associated more likely with anosmia and ageusia [11]. During SARS-CoV-2 infection, some patients refer the following: conjunctivitis (N = 9), visual acuity reduction (N = 6), rotatory vertigo (N = 6), tinnitus (N = 5) [11].

In a cross-sectional study Membrilla et al. named headache as a cardinal symptom of COVID-19 disease. Of 145 confirmed and probable COVID-19 patients, 99 (68.3%) reported headaches. A total of 54/99 (54.5%) were classified with probable COVID-19 and 45/99 (45.5%) with confirmed COVID-19. Once more, a great number of patients (86% patients) were observed with bilateral headache. The most frequent localizations were frontal or whole head; every localization was present in 34% patients. 89% patients described headache intensity >5 in the VAS scale, 60% of them described their headache as >7 points in the VAS scale. Headaches existing longer than 24 h without remission affected 45% patients [12].

The European clinical presentation appears different from that reported in Asia. According to recent studies, the COVID-19 infection of both hospitalized and nonhospitalized patients in Asia was mainly associated with fever, cough, dyspnoea and fatigue [5,6,7,8,9,13,14]. In European studies, more common reported symptoms were connected with headaches, olfactory and gustatory dysfunction, and gastrointestinal disturbances [11,13].

There could be some potential explanation of this phenomenon: the Asian studies only included hospitalized patients who were probably more affected by the disease (moderate-to-severe patients) in which pulmonary symptoms would be more prevalent. Another difference could be in the genetic pattern, based on mutations of the coronavirus in the Asian and European regions.

Headache features presented by patients were analyzed according to the third edition of the international classification of headache disorders (ICHD-3) [15].

We found that migraine visual aura preceding headache was noticed in 6/83 (7%) patients, unilateral headache character occurred in 6%, and the pulsating headache type without unilateral character was in 17% of participants. In greater frequency, nausea, vomiting, photo- and phonophobia, and escalation of physical effort were reported, although not fulfilling in these participants another migraine headache criteria for migraine without aura. Migraine-like headaches have already been described in association with viral infections [16,17]. In our study the headache associated with COVID-19 was rarely of a migraine phenotype. It did not fulfill complete ICHD-3 criteria for migraine without aura, and none of the participants had at least 5 migraine headache attacks.

We also found that tension-type-like headache character was reported in 43%, while pulsating quality was reported in 17%, accompanying nausea was reported in 31%, photo- and photophobia was reported in 14%, and escalation by physical activity was reported in 47% of our participants. Criteria include at least 10 headache episodes in infrequent and frequent TTH, headaches occurring on ≥15 days/month on average for >3 months in chronic TTH [15], which was not completed in this study. Probable TTH could be taken into consideration in some cases as possible diagnosis. According to headaches attributed to systemic viral infection criteria, it should have a diffuse character involving whole head, as was reported by 49% participants reporting headaches. Systemic viral infection was confirmed in the study by COVID-19 tests, there was no evidence of encephalitic involvement. Symptoms like neck stiffness, altered level of consciousness, seizures, and meningeal or focal signs could raise suspicions of CNS infection, and none of our patients presented them.

The symptoms developed in temporal relation to onset of SARS-CoV-2 infection and improved with improvement of the systemic viral infection. Diagnosis of headache attributed to systemic viral infection did not define duration of headache [15]. Some publications indicate atypical phenomenology and course of migraine attacks which can be observed during COVID-19 infection. One case report suggests that coronavirus may affect the bioelectrical activity of the brain, and perhaps by increasing activity, especially in the occipital lobes it may be responsible for CSD and the successive appearance of auras [18]. Nevertheless, these reports are casuistic.

During headache in COVID-19, analysis we should also consider headache in acute rhinosinusitis (ARS), because one headache type can mimic the other and ARS is a cause of one of the most prevalent secondary headaches. ARS is symptomatic of inflammation of the paranasal sinuses and nasal cavity mucosa. The criteria for the diagnosis established in 1997 by the Rhinosinusitis Task Force and divided symptoms in two major and minor groups are:Major: purulence on nasal examination, nasal obstruction, nasal discharge, hyposmia/anosmia, fever, facial pain/pressure, facial fullness; andMinor: ear pain/fullness, cough, dental pain, fatigue, halitosis, fever, headache.

For syndrome identification, we need either two major factors or one major plus two minor factors [19,20]. Headaches typical for ARS are classified in ICHD-3 criteria: develop in temporal relation to the onset of rhinosinusitis, getting worse or improved with worsening or improvement of the ARS, and exacerbated by pressure applied over the paranasal sinuses or, in the case of a unilateral rhinosinusitis headache, is ipsilateral to it. Gathering all these criteria and fact that 6% of participants in our study had unilateral headache, 7% admitted sinusitis symptoms, 28% had anosmia/hyposmia, and 29% had cough, it is more difficult to make an unambiguous diagnosis, so in every case a PCR laboratory test or rapid antigen test should be performed.

In our COVID-19 headache considerations and this extraordinary pandemic situation we should not forget old and well-known diseases which patients had before or diagnosed during SARS-CoV-2 pandemic. Many authors highlight that chronic diseases are always present in a certain percentage among the population, and lockdown, due to the COVID-19 pandemic, has caused significant disruption in oncological diagnostics and health-care management, especially when patients need long-term medical care and follow up visits. Montemurro N. focused on the problem of maintaining mostly life threatening operations in hospitals, elective surgeries in many hospitals all over the world have been cancelled to ensure adequate hospital capacity to respond to COVID-19 and to expand their intensive care capacity. Some elective and most nonelective surgeries must continue throughout any pandemic, as “old diseases” continue to exist, and non-COVID-19 patients require to be treated for other neurosurgical, neurological and psychiatric pathologies [21]. Trying to rescue coronavirus-infected patients, we should not forget about patients who needed our help from other well-known diseases.

## 5. Conclusions

This study is a thoroughgoing analysis of headache features related to COVID-19 disease. The frequency of headaches observed in the present study was higher than reported in many other studies mentioned before. It is possible that the frequency of headache was underestimated in other studies because of the respiratory symptoms for which patients were most frequently admitted to hospital.

COVID-19 related headaches do not fulfill the migraine without aura and tension-type headache criteria, they are mostly bilateral, tension-type-like, with intensity from middle to severe, sometimes lasting many days. It often escalates from physical effort. This is the first kind of attempt of such comparative characterization of headache in COVID-19 with a proposal for further investigations, in order to propose new type of headache to the International Classification of Headache Disorder: headache in SARS-CoV-2 infection, due to the fact that this headache has some features from other headaches: TTH, migraine and headache attributed to viral infection.

### Limitation of the Study

This study assessed symptoms of patients with mild forms of the disease, who did not need to be hospitalized. Neuroimaging, cerebrospinal fluid tests or ophthalmoscopy were not performed to rule out other causes of secondary headaches such as meningitis, encephalitis, and cerebrovascular diseases, which may be complications of COVID-19. None of our patients presented neck stiffness, altered level of consciousness, seizures, or had any meningeal or focal signs or other symptoms that raised the suspicion of a CNS infection. The study gathered mostly middle-aged patients with average age 39.3. This study characterizes type of headache in coronavirus infection without finding statistical correlation, therefore this research has no implementation of control group.

## Figures and Tables

**Table 1 brainsci-11-01406-t001:** Most frequent symptoms reported by our study patients.

Symptoms in COVID-19	Number of Participants Reporting Symptom
Extraocular muscle movement pain	6 (6%)
Skeletal muscle pain, joint pain	50 (50%)
Anosmia/hyposmia	28 (28%)
Ageusia/dysgeusia	22 (22%)
Cough	29 (29%)
Dyspnea	10 (10%)
Sore throat	6 (6%)
Gastrointestinal disturbances (nausea/vomiting/diarrhea/constipation)	20 (20%)
Sinusitis symptoms	7 (7%)
Conjunctivitis	1 (1%)
Drowsiness/fatigue/weakness	49 (49%)
Fever	60 (60%)
Sensory disturbances (hyperesthesia, paresthesia)	5 (5%)
Headache	83 (83%)

**Table 2 brainsci-11-01406-t002:** Headache characteristics, accompanying features, and treatment responses of patients experiencing.

Headache Characteristics	Number of Participants Reporting Symptom
Tension-type headache (squeezing pain)	36 (43%)
Pulsating headache	14 (17%)
Unilateral headache	5 (6%)
Whole head localization	41 (49%)
Frontal lobe localization	24 (29%)
Occipital lobe localization	8 (10%)
Length of headache episode	From 1h headache to 7 day headache; mean episode length 7 h 53 min.
Accompanying nausea/vomiting	26 (31%)
Accompanying photophobia/phonophobia	12 (14%)
Escalation by physical effort	39 (47%)
Migraine aura preceding headache	6 (7%)

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
