# Peer review of "Headache Attributed to SARS-CoV-2 Infection or COVID-19 Related Headache—Not Migraine-like Problem-Original Research"

_brainsci, 2021, doi:10.3390/brainsci11111406_

Round 1
Reviewer 1 Report
In this study, the authors collected 100 randomly selected healthcare workers with COVID-19 disease-related symptoms through a questionnaire study, and found that the headaches reported by these patients did not meet the criteria for migraine and tension-type headache. This study is somewhat interesting. However, I have some concerns about this study.
- The description of Table 2 is missing in the results.
- Some references are missing.
- According to descriptions in the manuscript, participants had at least 5 migraine attacks before being diagnosed with migraine, but the data provided in the study did not show the number of migraine attacks.
- The authors mentioned that the diagnosis of tension-type headache was not completed in this study, but concluded that the COVID-19-related headaches reported by these patients did not meet the criteria for migraine and tension-type headache.
- The authors described in the discussion that “In 1420 patients in European study the most prevalent symptoms were headache (70.3%), loss of smell (70.2%), nasal obstruction (67.8%), cough (63.2%), asthenia (63.3%), myalgia (62.5%), rhinorrhea (60.1%), gustatory dysfunction (54.2%) and sore throat (52.9%). Fever was reported by on 45.4%. Some patients added the following: conjunctivitis (N = 9), visual acuity reduction (N = 6), rotatory vertigo (N = 6), tinnitus (N = 5)[7,8]", this description is exactly the same as in Ref.7, please modify accordingly.
- The discussion mentioned "One European study focused on epidemio-logical characteristics of 1420 patients hospitalized with COVID-19 reported occurrence of headache in 998 patients (70%)[7]", which is a repetitive description in the previous paragraph.
- The authors mentioned "The frequency of headache observed in the present study was higher than reported in many other studies mentioned before", how did the authors come to the conclusion? What statistical methods were used in this study?
Author Response
Dear Reviewer,
Thank you very much for reviewing our manuscript and for all your suggestions! We did our best to adjust the article according to the instructions. All changes in article are in red color.
- Table 2 description misses the results. We rebuilt the Table 2 to join informations about headache localizations, we had asked participants in questionnaire: whole head headache, frontal lobe headache, occipital lobe headache. Now Table 2 description is coherent with Results.
- Lack of some references. We checked the article once again and added also new references: "Over the years, several different respiratory viruses have been shown to be able to penetrate the CNS (neuroinvasion). Most of them can infect neurons and glial cells (neurotropism) and result in the induction of neurological diseases[1]. However, it has now been proven indubitably that coronaviruses possess neurotropic and neuroinvasive properties in various hosts including humans, rats, pigs, rodents and fowls[2]. Coronaviruses first target respiratory and mucosal surfaces and then, depending of host and virus strain, may spread to other tissues (brain, eyes, liver, kidneys and spleen) and cause a range of pathologies such as pneumonia, encephalitis, neurodegenerative demyelination, hepatitis, enteritis, and nephritis among others[3]." Page 1 Line 19, as well in the last paragraph in Discussion part.
- Number of migraine attacks. We included participants into study with no past history, diagnosis of migraine headache. In Research Questionnaire were gathered questions about number and length of headache episodes in COVID-19 disease. On the basis of questionnaire answers none of participants had at least 5 migraine headache attacks during SARS-CoV-2 infection. So nobody of our group meet the ICHD-3 criteria for migraine without aura diagnosis.
- Criteria of TTH. TTH was diagnosed according ICHD-3 (Lasting from 30 minutes to seven days, bilateral location, pressing or tightening (non-pulsating) quality, mild or moderate intensity, not aggravated by routine physical activity, such as walking or climbing stairs, no nausea or vomiting, no more than one of photophobia or phonophobia).
- Double description. Citied description from Discussion part is modified, coherent in order and meaning with sentence citied from the same research (now this is Reference No 10).
- Repetitive description. We modified this statement "One European study focused on epidemiological characteristics of 1420 patients hospitalized with COVID-19 reported occurrence of headache in 998 patients (70%)" and placed it to the right part of the previous paragraph not to make repetitive description in 2 paragraphs. Page 4 Line 12.
- Frequency of headache/statistics. The aim of our research was not to find statistical correlation between headache and other symptoms in COVID-19, but attempt to characterize type of headache in this coronavirus infection. There were some suggestions that headache in COVID-19 can be TTH-like or migraine-like, we wanted to investigate these reports. We plan to continue research and ask participants about post-COVID symptoms, current research consist of preliminary results.
Kind regards,
Aleksandra Kacprzak
Izabela Domitrz
Daniel Malczewski
Reviewer 2 Report
Interesting, but look at these points to improve:
- Introduction section should be improved as well as the aim of the paper should be exposed better.
- "The aim of our research was not to find statistical correlation between
headache and other symptoms in COVID-19, but attempt to characterize type of headache in coronavirus infection" must be explain better. - "The disease can range from mild with asymptomatic or non-specific symptoms to severe with respiratory distress as in Tab. 1." Please report in the text at least the 3 more common symptoms in table 1.
- In the discussion section "According to recent studies, the COVID-19 infection of both hospitalized.. " The need to diagnose without forgetting the old disease should be highlighted. Please consider these 2 papers: Intracranial hemorrhage and COVID-19, but please do not forget "old diseases" and elective surgery. Brain Behav Immun. 2021 Feb;92:207-208. doi: 10.1016/j.bbi.2020.11.034. --- Prevalence of Anxiety, Depression, and Stress among Teachers during the COVID-19 Pandemic: A Rapid Systematic Review with Meta-Analysis. Brain Sci. 2021 Sep 3;11(9):11
- The discussion is a bit confused. Try reviewing by following an organizational line of thinking.
- Paper has some limitations as for example the absence of a control group and this should be stated at the end of discussion or in the limitation of the study.
Author Response
Dear Reviewer,
Thank you very much for reviewing our manuscript and for all your suggestions! We did our best to adjust the article according to the instructions. All changes in article are in red color.
- Introduction improvement. We rebuilt the Introduction part and added new references: "Over the years, several different respiratory viruses have been shown to be able to penetrate the CNS (neuroinvasion). Most of them can infect neurons and glial cells (neurotropism) and result in the induction of neurological diseases[1]. However, it has now been proven indubitably that coronaviruses possess neurotropic and neuroinvasive properties in various hosts including humans, rats, pigs, rodents and fowls[2]. Coronaviruses first target respiratory and mucosal surfaces and then, depending of host and virus strain, may spread to other tissues (brain, eyes, liver, kidneys and spleen) and cause a range of pathologies such as pneumonia, encephalitis, neurodegenerative demyelination, hepatitis, enteritis, and nephritis among others[3]." Page 1 Line 19. Also aim of the research is better, wider clarify at the end of this part.
- Aim of research explanation. We developed in Introduction "This is a trial to gathering all research participants complaints during their coronavirus infection, comparing with other well-known and classified headaches, also one secondary headache connected with ARS, then try to categorize headache related to SARS-CoV-2 infection." Page 1 Line 43.
- Report 3 more common symptoms in the text. New sentence is added Results part: "Participants most frequently reported headache, fever, skeletal muscles and joints pain, feeling of drowsiness, weakness in COVID-19 disease." Page 2 Line 24.
- Diagnose without forgetting old diseases. We added new paragraph in Discussion part: "In our COVID-19 headache considerations and this extraordinary pandemic situation we should not forget of old and well-known diseases which patients had before or diagnosed during SARS-CoV-2 pandemic. Many authors highlight that chronic diseases are always present in a certain percentage among the population and lockdown, due to the COVID-19 pandemic, has caused significant disruption in oncological diagnostics and health-care management, especially when patient need long-term medical care and follow up visits. Montemurro N. focused on the problem of maintaining in hospitals mostly life threatening operations, elective surgery in many hospitals all over the world has been cancelled to ensure adequate hospital capacity to respond to COVID-19 and to expand their intensive care capacity. Some elective and most nonelective surgeries must continue throughout any pandemic, as “old diseases” continue to exist, and non-COVID-19 patients require to be treat also for other neurosurgical, neurological and psychiatric pathologies[19]. Trying to rescue coronavirus infected patients not to forget about patients needed our help from other well-known diseases." Page 5 Line 36.
- Line of thinking in Discussion. In Discussion part we changed order of cited above researchers observations/comments to keep this part more clear for readers. One new paragraph was added at the end of Discussion part.
- Absence of a control group. We added to Limitation of the Study: "Study characterizes type of headache in coronavirus infection without finding statistical correlation, therefore research has no implementation of control group." Page 6 Line 13.
Kind regards,
Aleksandra Kacprzak
Izabela Domitrz
Daniel Malczewski
Round 2
Reviewer 1 Report
The authors responded to all my concerns in a satisfactory way.
Reviewer 2 Report
Authors solved all my criticisms.
This manuscript is a resubmission of an earlier submission. The following is a list of the peer review reports and author responses from that submission.
Round 1
Reviewer 1 Report
In this study, the authors conducted a questionnaire survey of 100 randomly selected healthcare employees with COVID-19 disease-related symptoms. The authors found that headache caused by COVID-19 did not meet the criteria for migraine with/without aura and tension-type headache according to the International Classification of Headache Disorders. However, since the result description is not very clear, it is difficult to evaluate the importance of the findings based on the current manuscript. Please see the comments and questions listed below.
- Because no statistical analysis was conducted in the study, it is difficult to draw conclusions from the results.
- The authors described the findings of many other researchers in the results, which should be added to the discussion.
- The result description is inconsistent with the data in the table.
- The manuscript cites 16 references, but 17 references are attached. Please check it carefully.
- The authors cited other studies but did not include references.
Author Response
Thank you very much for all your suggestions! We did our best to adjust the article according to the instructions. All changes in article are in red color.
1. The aim of our research was not to find statistical correlation between headache and other symptoms in COVID-19, but attempt to characterize type of headache in this coronavirus infection. There were some suggestions that headache in COVID-19 can be TTH-like or migraine-like, we wanted to investigate these reports. We plan to continue research and ask participants about post-COVID symptoms, current research consist of preliminary results. We added "The aim of our research was not to find statistical correlation between headache and other symptoms in COVID-19, but attempt to characterize type of headache in this coronavirus infection." in Page 2 Line 14.
2. Findings from other researchers from 'Results' were removed to 'Discussion'.
3. We improved data appearance in the table and modified data in 'Results" to make it more transparent for readers.
4. Part "Limitation of the study" had included 17th reference - we canceled it from that part.
5. We added references to other researchers quoted studies and removed references from "Results" part, as it was suggested in 'Instructions for Authors'.
Reviewer 2 Report
Dear Authors,
- Introduction is too short and does not show the aim of the study. What what the aim of the study? Describe the reason and novelty of your study.
- I'm afraid that just a simple questionaire is nor enough to diagnose the type of headache. Why you created your own questionarie and not used many existing validated ones? You only focused on migraine and TTH, what about other types of headache? The regional inflammation - named acute rhinosinusitis (ARS) is a cause of one of the most prevalent secondary headaches. Headache caused
by ARS has its own criteria in ICHD-3 (11.5.1). Since the symptoms of COVID-19 allow in many cases to diagnose ARS, so headache in COVID-19 can be mostly
attributed to ARS. Unfortunately you did not write a single sentence about it.
- Did subjects give informed consent to participate in the study?
4.What was the exclusion and inclusion criteria?
- Do not cite other authors in results section! Please read Instructions for authors carefully! https://www.mdpi.com/journal/brainsci/instructions
- Please attache a blank version of used questionaire ( suplementary materials)
- Please include the data about prior diagnosis of headache disorder in your patients
- Regarding Covid-19 - what was the disesae severity? were the patients hospitalised? Were subjects screened for disease progression?
- What was an average time interval from Covid-19 diagnosis to questionaire assesment?
- Add statistics to find the correlation between headache and other Covid symptoms, ie if fever and cough showed significant association with
headache or its phenotype
- Do you think that headache in COVID-19 could also be related to direct viral
damage of the central or peripheral nervous system during infection ( meningitis, encephalitis or encephalopathy)??
- The Conclusion does not reflect results
please explain, where in the results section a confirmation of conclusion sentences may be found:
This study is first thoroughgoing analysis of headache features related to COVID-19
COVID-19 related headache do not fulfill migraine without aura and tension-type headache criteria,
It often escalates by physical effort or coughing.
Frequently start at the same time with other COVID-19 symptoms.?
Author Response
Thank you very much for all your suggestions! We did our best to adjust the article according to the instructions. All changes in article are in red color.
1. Introduction is corrected to be better focused on aim of the study, as follow "We created our study to focus on primary headache disorders and its comparison to headache features in COVID-19, compare the results of other researchers, in order to diffuse and popularize that issue, make diagnostic way easier and try to categorize this new medical problem. The aim of our research was not to find statistical correlation between headache and other symptoms in COVID-19, but attempt to characterize type of headache in this coronavirus infection." in Page 2 Line 12.
2. We created our questionnaire to focus on primary headache disorder and its comparison to headache features in COVID-19 described by our participants and worldwide, especially on these 2 most frequent types of headache - TTH and migraine. We planed to find out if there are some common features for COVID-19 and primary headache disorders. We added "We created our questionnaire to focus on primary headache disorder and its comparison to headache features in COVID-19" in Page 2, Line 11.
3. Participants were firstly asked (only verbally) for permission to take part in the questionnaire study and all accepted to answered our questions. No one had objections or was forced to take part in questionnaire. We didn't claim for permission to Research Ethics Committee.
4. Exclusion criteria: participants under 18 year old and above 70 year old; hospitalization because of COVID-19. Page 2 Line 25.
Inclusion criteria: no history of migraine in the past; lack of symptoms giving suspicion of CNS infection. Page 2 Line 25.
5. We removed citations from 'Results' part.
6. Blank version of questionnaire, translated in English, will be included (in supplementary materials).
7. The question of prior headache disorders (only TTH and migraine) was included in questionnaire. We excluded participants only with migraine history before COVID-19.
8. None of our participants needed hospitalization because of COVID-19 disease - it was one of inclusion criteria. We gathered data for questionnaire in hospital where our participants work, they were not hospitalized at that place and any other. This information was also concluded in 'Limitation of the study'. Severity of their disease was mild or moderate, patients with COVID-19 are considered to have severe illness if they have SpO2 <94% and need oxygen therapy. We plan to continue research and ask participants about post-COVID symptoms, current research consist of preliminary results. We added "None of our participants needed hospitalization because of COVID-19 disease" Page 2 Line 31.
9. We didn't measure exact time interval between COVID-19 and questionnaire assessment. We only included into research participants who had COVID-19 in last 3 months from questionnaire data gathering. We added "Study included only participants who suffered from COVID-19 disease in last 3 months" in Page 2 Line 22.
10. The aim of our research was not to find statistical correlation between headache and other symptoms in COVID-19, but attempt to characterize type of headache in this coronavirus infection. There were some suggestions that headache in COVID-19 can be TTH-like or migraine-like, we wanted to investigate these reports. We added "The aim of our research was not to find statistical correlation between headache and other symptoms in COVID-19, but attempt to characterize type of headache in this coronavirus infection." in Page 2 Line 14.
11. Indeed, headache in COVID-19 can be related to viral damage of central nervous system in COVID-19, among others, because of complications like meningitis or encephalitis, which are itself often related to headache. This relation is suggested in many current research articles in PubMed. We added "Symptoms like neck stiffness, altered level of consciousness, seizures, meningeal or focal signs could raise suspicions of CNS infection and none of our patients presented them." in Page 5 Line 13.
12. We improved conclusions, now better confirm results.

Round 2
Reviewer 1 Report
The authors have addressed my concerns. I have no further questions. Thank you.
Author Response
We read the article precisely in order to find any stylistic or language mistakes and improved everything what was uncertainly written.
Thank you very much for reviewing our manuscript.
Kind regards,
Aleksandra Kacprzak
Daniel Malczewski
Izabela Domitrz
Reviewer 2 Report
Dear authors, you din not meet my all expectations.
Author Response
Dear Reviewer,
- Introduction development. We developed 'Introduction' part. We added: "According to studies, more than 35% of COVID-19 patients develop neurological symptoms. Some COVID-19 patients may present neurologic symptoms as the initial presentations of the disease [2]. There are some studies and reviews showing that the most common neurological symptom is headache, often accompanied by high fever,. Headache can occasionally be seen alone as the first sign of the disease [3]." Page 2 Line 5. We added also at the end of 'Introduction': "We also pay attention to possible similarity of headache during COVID-19 disease and headache in Acute Rhinosinusitis to avoid misdiagnose."Page 2 Line 23.
- Research design. We managed to provide in each main research paragraph specific introduction part, paragraph development with examples from other studies (not providing references in 'Results' and 'Conclusions' part), also kind of summary at the end of paragraph.
- Methods description. Inclusion and exclusion criteria are better written to be more clear for readers: "Study included only participants who suffered from COVID-19 disease in last 3 months and (...) We included in investigation only adult participants, the youngest 18-year old and the oldest 70-year old, with no history of hospitalization need because of COVID-19 and without symptoms making suspicion of CNS infection. Surveys completed by participants under age of 18, with past migraine headache history or those reporting inconsistent responses were excluded." Page 2 Line 27.
- Results presentation. Results are presented in tables with % share. They are now more clearly described in paragraph. We also added "Our participants also reported very diversified headache episode duration, often terminated by anti-inflammatory drugs or paracetamol, which more as half of participants administered during COVID-19 disease. Mean headache episode lasted 7h 53 minutes. Participants admitted persisting headache mostly, improving or ending by medications, in a lesser degree episodic." Page 3 Line 18
- Conclusions supported by results. Conclusions summarize the results and gives now open gate for other researchers to investigate and consider COVID-19 related headache. Also 'Conclusions' in Abstract are boosted now: "Headache accompanied COVID-19 disease did not fulfill all criteria for migraine with/without aura, tension-type headache, headache attributed to viral infection, also headache in ARS according to ICHD-3, but connect features from each of these headaches. More studies engaging numerous groups of participants should be conducted for better subject recognition."
We read the article precisely in order to find any stylistic or language mistakes and improved everything what was uncertainly written.
Thank you very much for reviewing our manuscript.
Kind regards,
Aleksandra Kacprzak
Daniel Malczewski
Izabela Domitrz
